# Constrained portfolio optimization with discrete variables: An algorithmic method based on dynamic programming

**Fereshteh Vaezi Jezeie** *, **Seyed Jafar Sadjadi, Ahmad Makui**

Department of Industrial Engineering, Iran University of Science and Technology, Narmak, Tehran, Iran

* f.vaezi.92@gmail.com

**Data Availability Statement:** All relevant data are within the paper and its Supporting Information files.

## Abstract

Portfolio optimization is one of the most important issues in financial markets. In this regard, the more realistic are assumptions and conditions of modelling to portfolio optimization into financial markets, the more reliable results will be obtained. This paper studies the knapsack-based portfolio optimization problem that involves discrete variables. This model has two very important features; achieving the optimal number of shares as an integer and with masterly efficiency in portfolio optimization for high priced stocks. These features have added some real aspects of financial markets to the model and distinguish them from other previous models. Our contribution is that we present an algorithm based on dynamic programming to solve the portfolio selection model based on the knapsack problem, which is in contrast to the existing literature. Then, to show the applicability and validity of the proposed dynamic programming algorithm, two case studies of the US stock exchange are analyzed.

## 1.Introduction

Portfolio optimization is one of the most important issues in financial markets and has many applications in financial planning and decision making. In the early 1950s, the theoretical foundations of this issue and the modern portfolio theory were founded by Markowitz [1,2]. Traditional asset allocation methods like Markowitz theorem gives the solution as a percentage and this ratio may suggest allocation of half of a share on the market, which is impractical. Many portfolio optimization problems deal with allocation of assets which carry a relatively high market price. Therefore, it is necessary to determine the integer value of assets when we deal with portfolio optimization. Therefore, Vaezi et al. [3–5] study the portfolio optimization problem with integer variables. The Genetic Algorithm(GA) and Discrete Firefly Algorithm (DFA) is designed to find the near optional solutions of knapsack-based portfolio optimization with discrete variables. The aim of this study is to find a more efficient method for the portfolio optimization problems with discreet variables. Hence, in this study, an algorithmic method based on dynamic programming is designed for the knapsack-based constrained portfolio optimization to find the optional solutions, exactly.

Dynamic programming is an efficient mathematical method for multi-stage optimization problems. Dynamic planning, using systems-oriented processes based on the two properties of

**Funding:** The authors received no specific funding for this work.

**Competing interests:** The authors have declared that no competing interests exist.

overlapping sub-problems and optimal infrastructure, provides a combination of sequential decisions that maximize computational efficiency. Bellman [6] introduced this method in 1953. Unlike linear programming, there is no standard framework for formulizing dynamic programming problems. In fact, dynamic programming provides a general approach to solve these types of problems. In each case, special equations and mathematical relations must be written that match the conditions of the problem. The main contribution to multi-stage financial decisions has been made by Merton [7,8]. AÏT-SAHALI and Brandt [9] also used this template based on dynamic programming in continuous time. Sadjadi et al. [10] proposed a dynamic programming approach for the classical mean-variance Markowitz model, which included risky and riskless assets. Thus, Zhu et al. [11] presented a generalized mean-variance portfolio optimization with the possibility of controlling bankruptcy risk using the Lagrangian dual method and the parametric approach of the dynamic programming. Shin [12] presented the optimal consumption and portfolio selection model with risky and riskless assets using the dynamic programming based on the approach presented by Karatzas et al. [13] to extract the value function and optimal strategies in closed form. Using Bi-level programming and dynamic programming techniques, Chen and Song [14] examined a multi-period portfolio optimization by considering and controlling bankruptcy risk in the financial market. Moreover, some studies have used a dynamic programming approach based on the Hamilton-Jacobi-Bellman equations to examine limited portfolio optimization in the probability or expected size of wealth or consumption [15–19]. Palczewski [20] suggested an efficient numerical algorithm based on Bellman's dynamic programming approach to optimize a dynamic portfolio optimization with transaction costs that depend on different times and scenarios. Najafi and Pourahmadi [21] proposed a heuristic method based on dynamic programming for a multi-period portfolio optimization problem with transaction costs under uncertain conditions. Moreover, Guigues [22–24] suggested dual dynamic programming with cut selection for convex optimization problems and proved that the obtained method is faster than methods such as simplex in portfolio optimization problems. Mohebbi and Najafi [25] considered a multi-period horizon and transaction costs in the proposed model. They used a median absolute deviation as a risk measurement. Moreover, the uncertain condition was also considered based on the scenario tree in this model and dynamic programming was applied to solve this model. Moreover, Cui et al. [26] analyzed management fees in portfolio optimization using stochastic dynamic programming and Valladão et al. [27] provided a Markov chain stochastic dual dynamic programming in order to assign the asset allocation. As well as, Zhang et al. [28] combined the approximate dynamic programming method with the game theory and create an iteration method to apply the optimal portfolio. Hence, some studies have concentrated on the project portfolio optimization problem by applying stochastic dynamic programming such as Nowak and Trzaskalik [29].

As noted above, the mentioned methods based on dynamic programming are presented to solve classical portfolio optimization problems with continuous variables. Given that capital markets are facing a wider range of investors day by day and investors have very little time to choose the appropriate shares, any delay in decision-making can lead to a reduction in investor returns. Therefore, in this study, we are focused to propose an algorithmic method based on dynamic programming to solve the knapsack-based constrained portfolio optimization with discrete variables.

The proposed algorithm based on dynamic programming divides complex large-scale portfolio optimization problems into simpler sub-problems and stores their results to prevent recalculation of results. The proposed algorithm based on dynamic programming finds the optimal solution for the sub-problems and then makes an informed choice by combining the results of these sub-problems to find the most optimal solution. In fact, in the proposed

algorithm, instead of calculating the same sub-problems over and over again, the solutions are stored in the first memory and used in the next steps if needed. In the proposed algorithm, this feature is a great idea to save time and memory, where using the extra space, the time required to find the optimal solution is improved. According to the comparison of the results of solving the proposed algorithm based on dynamic programming with the results of solving the meta-heuristic algorithms, the proposed algorithm has the ability to achieve the most optimal solution in an acceptable time. Moreover, the greedy algorithm seeks an optimal solution at every local stage. Thus, greedy algorithms can offer a hypothesis that seems optimal at the time, but looks worrying in the future and does not guarantee global optimization.

The outline of the paper is as follows. According to the knapsack problem, the portfolio selection model is presented in Section 2. In Section 3, a proposed algorithm based on dynamic programming is designed for the knapsack-based constrained portfolio optimization. In section 4, two case studies of the US stock exchange are analyzed. In Section 5, the results of the suggested algorithm are compared with another method to show the efficiency of the proposed techniques. Finally, a summary of the paper and some results are provided in Section 6.

## 2. Portfolio optimization problem

The portfolio selection model is based on the knapsack problem and the following notations are applied:

$E(\mathcal{R}_P)$ the expected return of a portfolio;

Z the objective function with the coefficient $(\lambda)$ as a penalty;

$\lambda$ the coefficient of the cardinality constraint;

$N$, the dimension of the decision variable;

$B$, the budget;

$\mathcal{R}_i$, the rate of return per unit of share $i$;

$\rho_i$, the price per unit of share $i$;

$l_i$, the lower bound of share $i$;

$u_i$, the upper bound of share $i$;

$k$, the number of $y_i$ that can be in the portfolio;

$x_i$, the integer variable that indicates the number of share $i$;

$y_i$, the binary variable; $y_i = 1$, if share $i$ is involved in the portfolio, and $y_i = 0$ in any other way.

The proposed model is formulated as follows:

$$\max E(\mathcal{R}_P) = \sum_{i=1}^{N} \mathcal{R}_i x_i \tag{1}$$

S. t.

$$\sum_{i=1}^{N} \rho_i x_i \le B, \tag{2}$$

$$\sum_{i=1}^{N} y_i = k, \tag{3}$$

$$l_i y_i \le x_i \le u_i y_i, \forall i \in \{1, 2, \ldots, N\} \tag{4}$$

$$y_i \in (0, 1), \forall i \in \{1, 2, \ldots, N\}, \tag{5}$$

$$x_i \in int, \forall i \in \{1, 2, \ldots, N\}. \tag{6}$$

Eq (1) provides the portfolio that has the highest returns with respect to a consent of the

considered constraints. Eqs (2), (3) and (4) are the budget, cardinality, and quantity constraints, respectively. This model, which has discrete variables (Eqs (5) and (6)), has two very important features; achieving the optimal number of shares as an integer and with masterly efficiency in portfolio optimization for high priced stocks. These features have added some real aspects of financial markets to the model and distinguish them from other previous models (Vaezi et al. [3–5]).

As the number of constraints in the portfolio selection models increases, the complexity of implementing the dynamic programming approach increases and sometimes makes the process difficult or impossible. We have tried to simplify this process by adding a cardinality constraint with a coefficient to the objective function. Therefore, the previous model is rewritten as follows:

$$Z = \max \sum_{i=1}^{N} \mathcal{R}_i x_i - \lambda \left( \sum_{i=1}^{N} y_i \right) \tag{7}$$

S.t.

$$\sum_{i=1}^{N} \rho_i x_i \leq B, \tag{8}$$

$$l_i y_i \leq x_i \leq u_i y_i, \forall i \in \{1, 2, \ldots, N\} \tag{9}$$

$$y_i \in (0, 1), \forall i \in \{1, 2, \ldots, N\}, \tag{10}$$

$$x_i \in int, \forall i \in \{1, 2, \ldots, N\}. \tag{11}$$

In fact, Eq (3) is added to the objective function with the coefficient ($\lambda$) as a penalty. The important point is that the objective function becomes non-convex in this case.

## 3. Proposed algorithm based on dynamic programming

Dynamic programming is generally a powerful technique for algorithm design and can be considered as a kind of exhaustive search. Dynamic programming divides a complex problem into sub-problems and stores their results to prevent recalculation of results. Instead of calculating the same sub-problem over and over again, it is necessary to save your solution in the first memory and use it in the next steps if needed. This is a great idea to save time and memory, where it is accessible to use the extra space in order to improve the time required to find a solution [30].

An important feature of problems that can be solved by dynamic programming is interference. This feature distinguishes dynamic programming from other methods such as the divide and conquer technique [31].

In other methods such as greedy algorithms, the best option is only chosen by considering the existing conditions. Thus, greedy algorithms can provide a hypothesis that seems optimal at the time but does not guarantee a global optimization in the future. However, dynamic programming finds the optimal solution for the sub-problems and makes an informed choice to find the most optimal solution by combining the results of these sub-problems [32].

The first step to solve the problems using a dynamic programming approach is to determine the stage, state, and action for that problems. In the proposed dynamic programming approach, each stage belongs to a type of share or asset. Moreover, the state is the amount of available budget, which is defined as an interval. In fact, the problem shifts from continuous to discrete states with budget interruptions. Finally, the action is considered the number of shares. The number of shares fluctuates between their predetermined upper and lower bounds

and is an integer. The solution method is the backward recursion approach using the following recursive equation.

$$f(i, S_i, x_i) = \phi(i, S_i, x_i) + f^*(i+1, S_{i+1}), i \in \{1, 2, \ldots, N\}. \tag{12}$$

In Eq (12), $i$, $S_i$ and $x_i$ are the stage, state, and action, respectively. $\phi(i, S_i, x_i)$ is the value considered for decisions making based on three input parameters. In fact, this value is the rate of return obtained from $\mathcal{R}_i x_i - \lambda$. $\lambda$ is obtained by a sequential approximation method.

Note that, in addition, to calculate the rate of return, the amount of consumption budget resulting from this decision is also calculated according to $\rho_i x_i$ in each section. Thus, $f^*(i+1, S_{i+1})$ is the maximum profit or return of the stage $i+1$ for the remaining capital.

Given that the present problem has become a non-convex optimization problem, it is important to pay attention to the final table and select the global optimal point in the proposed approach. In the following, the pseudocode of the proposed dynamic programming algorithm is described in detail.

Step 1 describes the structure of dynamic scheduling tables. In this step, the solution method, the state, the number of rows, the actions, and the number of columns ($m_i$). The solution method is the backward recursion approach ($i = N, \ldots, 1$). Note that, the state depends on the length of the selected step ($L$) for the budget interval defined by the user. The states intervals are defined as $[S_{\Gamma-1}, S_\Gamma)$, $\Gamma \in \{1, 2, \ldots, n\}$ and according to $S_\Gamma = \Gamma^* L$ ($S_0 = S_1 = 0$, $S_n = B$). Therefore, the number of rows ($n$) is defined according to $n = \left[\frac{B}{L}\right]$. Moreover, the actions in stage $i$ are determined by the number of shares that fluctuate between their predetermined lower bound ($l_i$) and upper bound ($u_i$). Hence, the number of table columns in stage $i$ ($m_i$) is determined by the number of actions in stage $i$ according to $m_i = u_i - l_i + 1$. Hence, six variables with the symbols; $R_i$, $P_i$, $R_{i+1}^*$, $P_{i+1}$, $P_T$ and $R_T$ are being assigned in each cell. $R_i$ and $P_i$ are the return and price of the current stage (stage $i$) in each cell. $R_{i+1}^*$ and $P_{i+1}$ are the best return and its price of the pervious stage (stage $i+1$) in each cell. $P_T$ and $R_T$ are the total price and return in each cell. In addition, the last two columns contain the maximum values of the objective function in each row and the action assigned to it, respectively.

Step 2 describes how to fill each cell in the first state table. Moreover, Step 3 describes how to fill each cell in the next states tables. Step 4 also provides the final solution. Due to the objective function of the problem is non-convex, the final solution to the problem is to select the maximum value from the optimal column values associated with each row in the final table and the actions assigned to that value. Finally, the pseudocode of the proposed dynamic programming approach is as follows:

**Algorithm 1: Pseudocode of dynamic programming.**

*Step 1:* <u>Table shape</u>

```
    For i = N,...,1 then
    Set mᵢ = uᵢ−lᵢ+1 and j = 1,2,...,mᵢ
    Set n = [B/L] and Γ = 1,2,...,n
    Create a table with mᵢ columns and n rows and call it the i-th
window
    Put xᵢⱼ = lᵢ+(j−1) at the top of column j
    Assign the Γ-th row of the table to interval [Sᵧ₋₁,Sᵧ)
    If Γ = 1, then
    Set Sᵧ₋₁ = Sᵧ = 0
    End if
    If Γ<n, then
    Set Sᵧ = Γ*L
    End if
    If Γ = n, then
    Set Sₙ = B
```

```
End if
```
Display the point of the table where the j-*th* column and the Γ-*th* row meets $C_{\Gamma j}$.

*Assign six variables with the symbols*; $R_i, P_i, R_{i+1}^*, P_{i+1}, P_T$ *and* $R_T$ in each cell, which in the general view of the table we display them with the symbols; $C_{\Gamma j}.R_i, C_{\Gamma j}.P_i, C_{\Gamma j}.R_{i+1}^*, C_{\Gamma j}.P_{i+1}, C_{\Gamma j}.P_T$ *and* $C_{\Gamma j}.R_T$

Consider two columns for each table to store the value of $f^*(x_i)(,S_\Gamma), (\Gamma = 1, 2, \ldots, n)$ *and* $x_{\Gamma i}^*$

Assign two variables with the symbols; $R_i^*$ *and* $P_i$ in each cell of $f^*(x_i, S_\Gamma), (\Gamma = 1, 2, \ldots, n)$, which in the general view of the table we display them with the symbols; $f^*(x_i, S_\Gamma).R_i$ *and* $f^*(x_i, S_\Gamma).P_i$

**Step 2:** <u>*First state*</u>
```
If i = N then
```
Set $C_{\Gamma j}.R_{N+1} = 0 \,\forall\, \Gamma = 1, 2, \ldots, n.$ and $C_{\Gamma j}.P_{N+1} = 0 \,\forall j = 1, 2\ldots, m_i.$
```
For j = 1,2...,mi do
```
Set $w_{ij} = x_{ij}^* \rho_i$ and $r_{ij} = \mathcal{R}_i * x_{ij} + \lambda^*$
```
Find the interval which the wij belongs to it.
Suppose that wij belongs to k-th interval.
Set Ckj.Pi = wij and Ckj.Ri = rij
End for
For p = 1,2,...,n.
Find the maximum value among the Cpj. Ri∀ j = 1,2...,mi
```
Suppose $J = Max_{j=1,2\ldots,m_i} \{C_{pj}.R_i\}$

*Set* $f^*(x_i, S_p).R_i = C_{pJ}.R_i$ *and* $f^*(x_i, S_p).P_i = C_{pJ}.P_i$
```
Get the optimal vector Xpi* based on the optimal answer index
End for
End if
```
**Step 3:** <u>*Next states*</u>
```
If i<N then
Fore j = 1,2...,mi do
```
Set $w_{ij} = x_{ij}^* \rho_i$ and $r_{ij} = \mathcal{R}_i * x_{ij} + \lambda^*$
```
Find the interval which the wij belongs to it.
Suppose that wij belongs to k-th interval.
Set Ckj.Pi = wij and Ckj.Ri = rij
Set Zkj = Sk−wij
Go to i+1−th window
Find the interval which Zkj belongs to that
Suppose Zkj belongs to I−th interval
```
If $Z_{kj} \geq f^*(x_{i+1}, S_I).P_{i+1}$ then
Set $C_{kj}.R_{i+1}^* = f^*(x_{i+1}, S_I).R_{i+1}$ and $C_{kj}.P_{i+1} = f^*(x_{i+1}, S_I).P_{i+1}$
Else if $Z_{kj} < f^*(x_{i+1}, S_I).P_{i+1}$ then
Set $C_{kj}.R_{i+1}^* = f^*(x_{i+1}, S_{I-1}).R_{i+1}$ and $C_{kj}.P_{i+1} = f^*(x_{i+1}, S_{I-1}).P_{i+1}$
```
End if
```
Set $C_{kj}.R_T = C_{kj}.R_{i+1}^* + C_{kj}.R_i$ and $C_{kj}.P_T = C_{kj}.P_{i+1} + C_{kj}.P_i$
```
For p = 1,2,...,n. p≠k
```
Set $q_{pj} = S_p - w_{ij}$
```
Go to i+1−th window
Find the interval which qpj belongs to that
Suppose qpj belongs to I−th interval
```
If $q_{pj} > 0$ and $q_{pj} \geq f^*(x_{i+1}, S_I).P_{i+1}$ then
Set $C_{pj}.R_{i+1}^* = f^*(x_{i+1}, S_I).R_{i+1}$ and $C_{pj}.P_{i+1} = f^*(x_{i+1}, S_I).P_{i+1}$
Else if $q_{pj} > 0$ and $q_{pj} < f^*(x_{i+1}, S_I).P_{i+1}$ then
Set $C_{pj}.R_{i+1}^* = f^*(x_{i+1}, S_{I-1}).R_{i+1}$ and $C_{pj}.P_{i+1} = f^*(x_{i+1}, S_{I-1}).P_{i+1}$
Set $\xi = C_{pj}. P_{i+1} + w_{ij}$
```
If ξ∈p−th interval then
```

```
                Set $C_{pj}.P_i = w_{ij}$ and $C_{pj}.R_i = r_{ij}$
                Set $C_{pj}.P_T = C_{pj}.P_{i+1} + C_{pj}.P_i$ and $C_{pj}.R_T = C_{pj}.R^*_{i+1} + C_{pj}.R_i$
                Else if $\xi \notin p-th$ interval then
                Set $C_{pj}.P_i = 0$ and $C_{pj}.R_i = 0$
                Set $C_{pj}.R^*_{i+1} = f^*(x_{i+1}, S_p).R_{i+1}$ and $C_{pj}.P_{i+1} = f^*(x_{i+1}, S_p).P_{i+1}$
                Set $C_{pj}.P_T = C_{pj}.P_{i+1}$ and $C_{pj}.R_T = C_{pj}.R^*_{i+1}$
                End if
                Else if $q_{pj} < 0$ then
                Set $C_{pj}.P_i = 0$ and $C_{pj}.R_i = 0$
                Set $C_{pj}.R^*_{i+1} = f^*(x_{i+1}, S_p).R_{i+1}$ and $C_{pj}.P_{i+1} = f^*(x_{i+1}, S_p).P_{i+1}$
                Set $C_{pj}.P_T = C_{pj}.P_{i+1}$ and $C_{pj}.R_T = C_{pj}.R^*_{i+1}$
                End if
                End for
                End for
                For $p = 1,2,...,n$ then
                Find the maximum value among the $C_{pj}.R_T$ $\forall$ $j = 1,2...,m_i$
                Suppose $J = Max_{j=1,2...,m_i}\{C_{pj}.R_T\}$
                Set $f^*(x_i, S_p).R_i = C_{pJ}.R_T$ and $f^*(x_i, S_p).P_i = C_{pJ}.P_T$
                Get the optimal vector $X_{p_i}^*$ based on the optimal answer index
                End for
                End if
     Step 4: Last solution
                If $i = 1$ then
                Set $Sol = max_{p=1,2...,n} f^*(x_1, S_p)$
                Set $X^* = X^*_{Sol}$
                End if
                End for
```

Figs 1 and 2 show the flowchart of the proposed algorithm based on dynamic programming to better understand the suggested approach.

## 4.Numerical results

In this section, two case studies are presented. The first case study for the knapsack-based portfolio optimization problem is considered in small dimensions so that the proposed approach is better understood and the problem can be resolved manually to verify the validity of the results. The second case study for the knapsack-based portfolio optimization problem is considered in larger dimensions to further investigate the program execution time and sensitivity analysis of the proposed algorithm in each state.

### 4.1 Case study (1)

Table 1 provides the information on the prices ($\rho_i$), returns($\mathcal{R}_i$), lower bounds ($l_i$) and upper bounds ($u_i$) of shares belong to five environmental companies. To obtain this information, daily time series data during a five-year period from 1/9/2014 to 1/9/2019 has been used.

In the case study, the budget ($B$) and the step length ($L$) are 650\$ and 65, respectively. Therefore, the number of the state is 10. As well as, given that there are five different types of shares, five stages are defined in the case study. The actions are also considered the number of shares in each stage. Moreover, the number of $y_i$ that can be in the portfolio is considered 2 ($K = 2$) and the solution method is backwards.

The first step to solve the case study is to obtain the optimal value of the coefficient of the cardinality constraint ($\lambda$) that is $\lambda^*$. $\lambda^*$ is obtained from the sequential approximation method using the designed MATLAB code (S1 Appendix). Moreover, the Intlinprog solver in MATLAB has also been used to check the accuracy and validity of the designed MATLAB

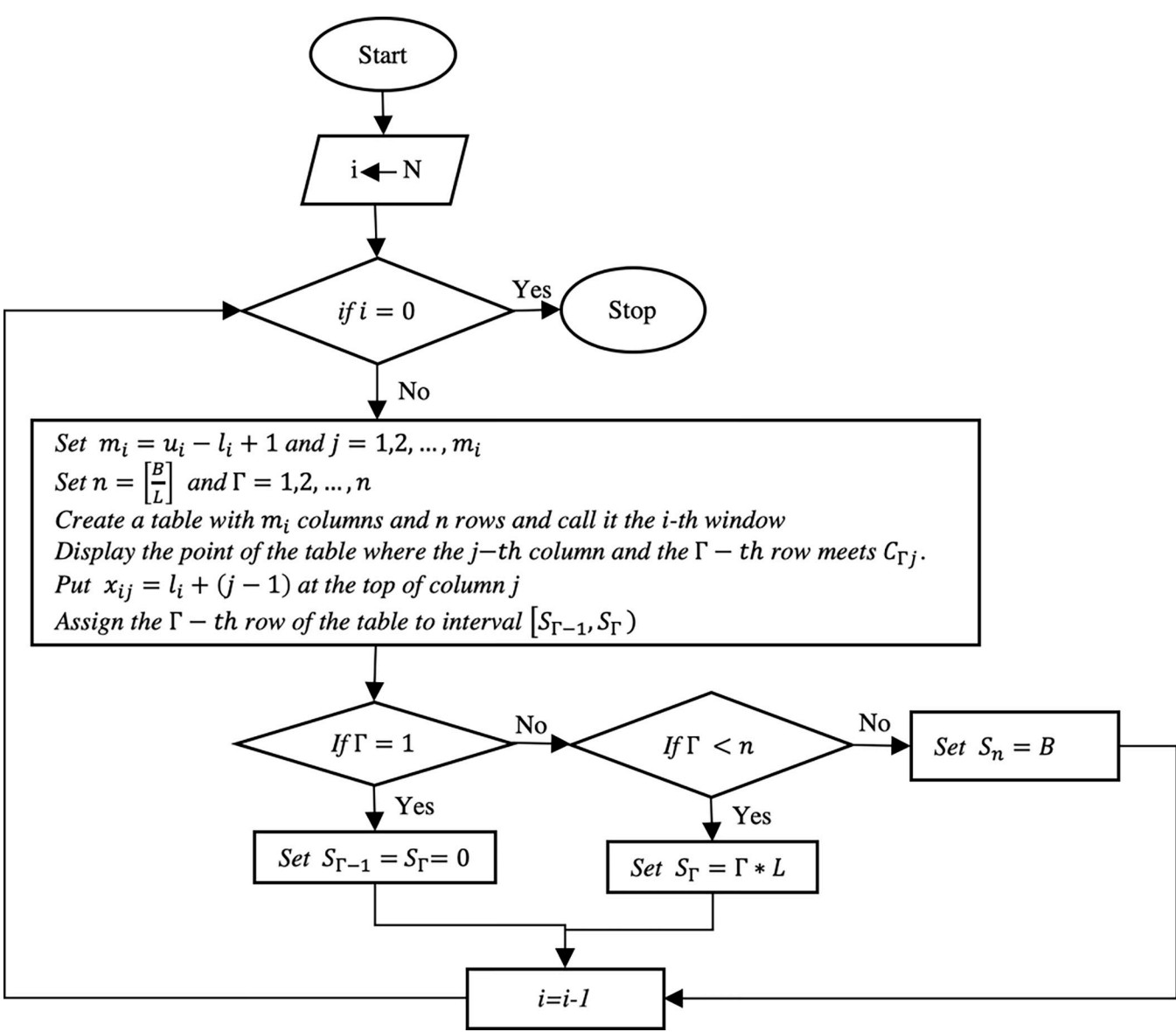

**Fig 1. Table shape.**

code results (S2 Appendix). Eventually, Information on the objective function and variables for different values of $\lambda$ in the interval $[-2\ 2]$ with a step length of 0.05 is provided in S3 Appendix. According to the results, the first place where $K$ in the cardinality constraint equals 2 ($K = 2$) is where $\lambda$ is -1.1($\lambda^* = -1.1$). Eventually, this model was solved by the proposed algorithm code based on dynamic programming provided in S4 Appendix. The software output is provided in S5 Appendix. The results include $y_1^* = 0$, $y_2^* = 1$, $y_3^* = 0$, $y_4^* = 1$, $y_5^* = 0$, $x_2^* = 7$, $x_4^* = 6$, $Z = 32.9346$ ($Z = \max \sum_{i=1}^{n} \mathcal{R}_i x_i - \lambda(\sum_{i=1}^{n} y_i)$) and $E(\mathcal{R}) = 30.7346$ takes 1.320s to precede. Moreover, to gain validity and reliability of the designed code, the case study is manually solved in the form of dynamic scheduling tables (S6 Appendix).

Table 2 shows the results of various changes in the state by changes in the step length ($L$) belonging to budget intervals and sensitivity analysis.

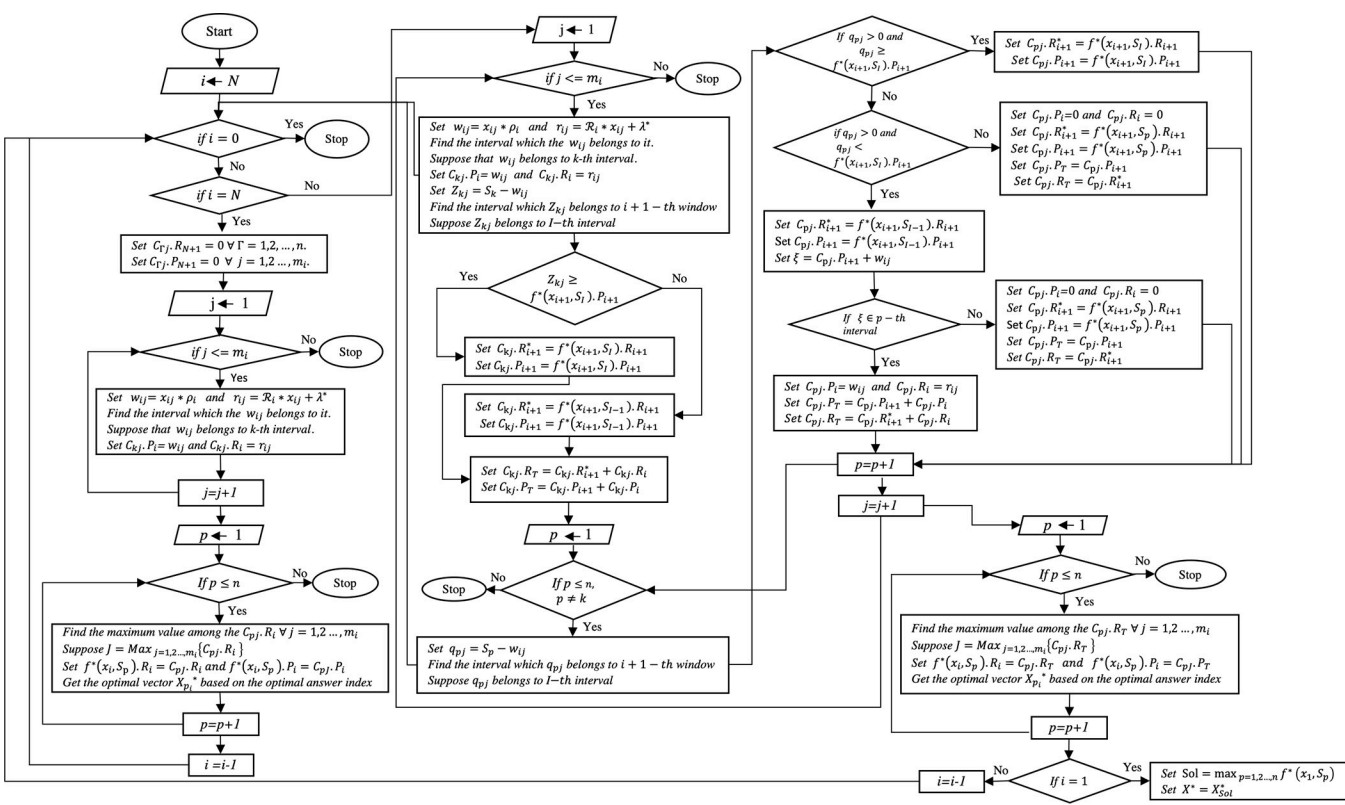

**Fig 2. Flowchart of the proposed algorithm.**

In this case study, the change of $L$ does not change the answer. In other words, the error rate of different changes in the state by changes in the step length ($L$) belongs to budget intervals is zero for this case study in the proposed dynamic programming approach.

## 4.2 Case study (2)

The second case study includes the Dow Jones Industrial Average listed on the New York Stock Exchange. The data includes daily time series over a period of 5 years from 1/9/2014 to 1/9/2019. Table 3 provides the information on the prices ($\rho_i$), returns($\mathcal{R}_i$), lower bounds ($l_i$) and upper bounds ($u_i$) of shares.

In the case study, the budget ($B$) and the number of $y_i$ that can be in the portfolio ($K$) are 3100\$ and 6, respectively. Table 4 provides information on the objective function and variables for different step lengths ($L$) belonging to budget intervals. Noted that each $L$ has a special $\lambda$, which is obtained with a sequential approximation method and step length of 0.05.

**Table 1. Information on environmental shares.**

|   | Symbol | $\rho_i$ | $R_i$ | $l_i$ | $u_i$ |
|---|--------|----------|-------|-------|-------|
| **1** | NEE | 138.2866 | 2.0009 | 2 | 6 |
| **2** | ECOL | 50.9708 | 2.8934 | 6 | 11 |
| **3** | WM | 72.5707 | 3.3595 | 5 | 9 |
| **4** | ORA | 48.5614 | 1.7468 | 5 | 11 |
| **5** | WCN | 57.5193 | 1.4793 | 5 | 10 |

**Table 2. Sensitivity analysis of environmental shares.**

| L | K | λ | $y_i^*, x_i^*$ | Z | E(R) | $T_{(s)}$ |
|---|---|---|---|---|---|---|
| 5 | 2 | -1.1 | $s_2 = 7, s_4 = 6$ | **32.9346** | 30.7346 | **11.156** |
| 10 | 2 | -1.1 | $s_2 = 7, s_4 = 6$ | **32.9346** | 30.7346 | **4.885** |
| 15 | 2 | -1.1 | $s_2 = 7, s_4 = 6$ | **32.9346** | 30.7346 | **3.762** |
| 20 | 2 | -1.1 | $s_2 = 7, s_4 = 6$ | **32.9346** | 30.7346 | **2.588** |
| 35 | 2 | -1.1 | $s_2 = 7, s_4 = 6$ | **32.9346** | 30.7346 | **1.941** |
| 50 | 2 | -1.1 | $s_2 = 7, s_4 = 6$ | **32.9346** | 30.7346 | **1.345** |
| 65 | 2 | -1.1 | $s_2 = 7, s_4 = 6$ | **32.9346** | 30.7346 | **1.320** |
| 100 | 2 | -1.1 | $s_2 = 7, s_4 = 6$ | **32.9346** | 30.7346 | **0.845** |

According to Table 4, the answers of $y_i^*$ are the same for different step lengths (L) belong to budget intervals and selected λ for each L. Therefore, there is no error in this part. However, there is a difference in the selected number of each selected share ($x_i^*$). Due to one of the factors that affect the objective function is $x_i^*$, the objective function also changes. This type of

**Table 3. Information on Dow Jones shares.**

|  | Symbol | $u_i$ | $l_i$ | $R_i$ | $\rho_i$ |
|---|---|---|---|---|---|
| $S_1$ | BA | 15 | 5 | 1.0872 | **224.6760** |
| $S_2$ | WBA | 25 | 10 | 2.7748 | **75.3938** |
| $S_3$ | MMM | 17 | 7 | 1.3920 | **183.1025** |
| $S_4$ | PG | 20 | 8 | 3.6375 | **86.8938** |
| $S_5$ | KO | 29 | 5 | 7.2013 | **44.3095** |
| $S_6$ | AAPL | 21 | 6 | 1.1563 | **145.0188** |
| $S_7$ | AXP | 25 | 8 | 1.7023 | **86.7325** |
| $S_8$ | UTX | 20 | 10 | 1.9853 | **115.0255** |
| $S_9$ | CVX | 30 | 9 | 3.6870 | **109.3446** |
| $S_{10}$ | JNJ | 17 | 8 | 2.2278 | **120.2702** |
| $S_{11}$ | NKE | 22 | 10 | 1.7581 | **62.1748** |
| $S_{12}$ | UNH | 17 | 7 | 0.7944 | **176.3504** |
| $S_{13}$ | MSFT | 23 | 8 | 2.8306 | **74.5001** |
| $S_{14}$ | IBM | 24 | 7 | 2.4800 | **151.2383** |
| $S_{15}$ | TRV | 17 | 8 | 1.8364 | **120.2462** |
| $S_{16}$ | MRK | 28 | 13 | 4.7896 | **62.7606** |
| $S_{17}$ | XOM | 30 | 6 | 4.5384 | **82.5577** |
| $S_{18}$ | WMT | 33 | 10 | 3.0755 | **81.7679** |
| $S_{19}$ | GS | 31 | 3 | 0.6580 | **204.6388** |
| $S_{20}$ | CAT | 19 | 9 | 2.8098 | **106.9702** |
| $S_{21}$ | V | 21 | 10 | 0.6549 | **101.0469** |
| $S_{22}$ | CSCO | 35 | 8 | 8.2878 | **35.4706** |
| $S_{23}$ | HD | 16 | 7 | 1.3595 | **150.6731** |
| $S_{24}$ | JPM | 14 | 2 | 2.8655 | **84.7972** |
| $S_{25}$ | PFE | 15 | 10 | 9.7546 | **35.4576** |
| $S_{26}$ | MCD | 23 | 6 | 1.9919 | **139.4249** |
| $S_{27}$ | VZ | 25 | 4 | 8.8849 | **50.6500** |
| $S_{28}$ | INTC | 25 | 7 | 6.9075 | **39.4024** |
| $S_{29}$ | DIS | 30 | 6 | 1.2969 | **106.9172** |

**Table 4. Sensitivity analysis of Dow Jones shares.**

| L | K | $\lambda$ | $y_i^*, x_i^*$ | Z | $T_{(s)}$ |
|---|---|---|---|---|---|
| 5 | 6 | -16.7 | $s_2 = 13, s_5 = 7$ $s_{22} = 21, s_{25} = 15$ $s_{27} = 4, s_{28} = 8$ | 597.8439 | **491.722** |
| 10 | 6 | -16.48 | $s_2 = 12, s_5 = 11$ $s_{22} = 16, s_{25} = 15$ $s_{27} = 5, s_{28} = 8$ | 590.0002 | **250.226** |
| 15 | 6 | -12.25 | $s_2 = 14, s_5 = 8$ $s_{22} = 16, s_{25} = 15$ $s_{27} = 4, s_{28} = 8$ | 539.6810 | **179.231** |
| 20 | 6 | -11.65 | $s_2 = 15, s_5 = 7$ $s_{22} = 16, s_{25} = 15$ $s_{27} = 4, s_{28} = 8$ | **531.6545** | **132.407** |
| 50 | 6 | -17.95 | $s_2 = 16, s_5 = 9$ $s_{22} = 13, s_{25} = 15$ $s_{27} = 4, s_{28} = 7$ | **554.8610** | **55.427** |
| 80 | 6 | -19.35 | $s_2 = 16, s_5 = 7$ $s_{22} = 14, s_{25} = 14$ $s_{27} = 4, s_{28} = 8$ | **554.2991** | **34.108** |
| 100 | 6 | -16.85 | $s_2 = 16, s_5 = 9$ $s_{22} = 14, s_{25} = 14$ $s_{27} = 4, s_{28} = 7$ | **546.7942** | **30.201** |
| 150 | 6 | -16.95 | $s_2 = 15, s_5 = 6$ $s_{22} = 17, s_{25} = 13$ $s_{27} = 6, s_{28} = 7$ | **555.8941** | **20.547** |

error sometimes occurs due to the transformation of the budget state from continuous to an interval in the proposed approach.

Eventually, according to the program execution time in each state and the results, we find that the suggested dynamic programming method is very efficient.

## 5. Comparative study

In the section, the results of the suggested algorithm based on dynamic programming compare with another method to show the efficiency of the proposed techniques. To compare the efficiency of the proposed techniques with well-known existing techniques, Discrete Firefly Algorithm (DFA) and Genetic Algorithm (GA) designed to solve portfolio optimization problem with discrete variables according to Vaezi et al. [3–5].

In addition, to select appropriate parameters, the Taguchi method is used for five parameters of DFA. Finally, parameters selected in this algorithm are attractiveness ($\beta_0 = 1$),

**Table 5. The average results of solving DFA and proposed algorithm and comparing them with each other.**

| | Mean Solution of $E(R_p)$ | Mean Time (s) |
|---|---|---|
| Mean result of proposed algorihm | 655.0135 | 149.2336 |
| Mean result of DFA | 747.9531 | 59.693 |
| SE. Mean | 46.5 | 44.8 |
| S.D. | 65.7 | 63.3 |
| P-Value* | 0.042 | 0.258 |

* denotes rejection of the hypothesis at the 0.01 level.

**Table 6. The average results of solving GA and proposed algorithm and comparing them with each other.**

|  | Mean Solution of $E(R_p)$ | Mean Time (s) |
|---|---|---|
| Mean result of proposed algorihm | 655.0135 | 149.2336 |
| Mean result of GA | 766.1489 | 48.977 |
| SE. Mean | 55.6 | 50.1 |
| S.D. | 78.6 | 70.9 |
| P-Value* | 0.050 | 0.298 |

* denotes rejection of the hypothesis at the 0.01 level.

randomization parameter ($\varphi = 0.2$), absorption coefficient ($\gamma = 1$), number of iterations (*Max-Generation = 500*), and number of fireflies (*Population = 29*).

The average results of DFA compare with the average results of the proposed algorithm based on dynamic programming and their results are reported in Table 5.

As can be seen in Table 5, the standard errors of mean (SE. Mean), standard deviations (S. D.) and probability values (P-value) are 46.5, 65.7 and 0.042, simultaneously.

Next, the appropriate parameters of GA are Number of iterations (*MaxIt = 500*), Population size (*nPop = 500*), Single point crossover whit probability *0.8* and Swap mutation for permutation with probability *0.2*. These parameters are also selected using the Taguchi method.

Moreover, the average results of GA compare with the average results of the proposed algorithm based on dynamic programming and their results are also reported in Table 6.

As can be seen in Table 6, the standard errors of mean (SE. Mean), standard deviations (S. D.) and probability values (P-value) are 55.6, 78.6 and 0.050, simultaneously.

Note that the mean result of proposed algorithm based on dynamic programming is more accurate than the mean result of GA and DFA and also the program execution time of the proposed algorithm is acceptable, the proposed algorithm based on dynamic programming is very valid. Finally, all results confirm the reliability and credibility of the proposed algorithm based on dynamic programming to solve the constrained portfolio optimization problem whit discreet variables.

# 6. Conclusion

In the paper, we have presented a dynamic programming algorithm to solve the portfolio selection model based on the knapsack problem. This model, which has discrete variables, has two very important features; achieving the optimal number of shares as an integer and with masterly efficiency in portfolio optimization for high priced stocks. These features have added some real aspects of financial markets to the model and distinguish them from other previous models. To solve the knapsack based portfolio selection model in larger and more complex dimensions by proposed algorithm based on dynamic programming, MATLAB code has been used. Then, to show the applicability and validity of the proposed dynamic programming algorithm, two case studies have been considered. The first case study consists of shares belong to five environmental companies. The case study was solved using designed MATLAB code. Then, it was solved manually in the form of dynamic programming tables. Finally, by comparing the results of the two approaches, the validity of the designed code was analyzed. Hence, the second case study, which was larger in dimensions and included the Dow Jones shares, was reviewed. Finally, the results of the suggested algorithm based on dynamic programming were compared with another method to show the efficiency of the proposed techniques. To compare the efficiency of the proposed techniques with well-known existing techniques, DFA and

GA were designed to solve portfolio optimization problem with discrete variables according to Vaezi et al. [3–5].

Note that the mean result of proposed algorithm based on dynamic programming was more accurate than the mean result of GA and DFA and also the program execution time of the proposed algorithm was acceptable, the proposed algorithm based on dynamic programming was very valid. Finally, all results confirmed the reliability and credibility of the proposed algorithm based on dynamic programming to solve the constrained portfolio optimization problem whit discreet variables. Eventually, according to the program execution time in each state, the results confirmed that the designed dynamic programming algorithm is absolutely valid.

Moreover, the presentation and development of a dynamic programming approach for the portfolio optimization problem based on the knapsack problem, considering investment risk, uncertainty conditions and other practical constraints, can be considered as examples of future research.

## Supporting information

**S1 Appendix. The designed MATLAB code for obtaining the optimal value of the coefficient of the cardinality constraint.**
(PDF)

**S2 Appendix. The Intlinprog solver in MATLAB.**
(PDF)

**S3 Appendix. Information on the objective function and variables for different values of $\lambda$.**
(PDF)

**S4 Appendix. The designed MATLAB code of the proposed algorithm based on dynamic programming.**
(PDF)

**S5 Appendix. The software output of the fifth environmental share.**
(PDF)

**S6 Appendix. The manually solution of the fifth environmental share.**
(PDF)

## Acknowledgments

We are also immensely grateful to the Editor-in-chief, Guest Editors and anonymous referees for sharing their insightful comments with us during the course of this research.

## Author Contributions

**Conceptualization:** Fereshteh Vaezi Jezeie.

**Data curation:** Fereshteh Vaezi Jezeie.

**Methodology:** Fereshteh Vaezi Jezeie.

**Software:** Fereshteh Vaezi Jezeie.

**Supervision:** Seyed Jafar Sadjadi, Ahmad Makui.

**Validation:** Seyed Jafar Sadjadi, Ahmad Makui.

**Visualization:** Seyed Jafar Sadjadi, Ahmad Makui.

**Writing – original draft:** Fereshteh Vaezi Jezeie.

**Writing – review & editing:** Fereshteh Vaezi Jezeie.

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
