## [Decision Letter · Decision Letter 0]

22 Apr 2022

PONE-D-22-09775A new dynamic programming algorithm to constrained portfolio optimization with discrete variablesPLOS ONE

Dear Dr. Jezeie,

Thank you for submitting your manuscript to PLOS ONE. After careful consideration, we feel that it has merit but does not fully meet PLOS ONE’s publication criteria as it currently stands. Therefore, we invite you to submit a revised version of the manuscript that addresses the points raised during the review process.

We look forward to receiving your revised manuscript.

Kind regards,

Seyedali Mirjalili

Academic Editor

PLOS ONE

Journal Requirements:

"No"

Reviewers' comments:

Reviewer's Responses to Questions

**Comments to the Author**

1. Is the manuscript technically sound, and do the data support the conclusions?

Reviewer #1: No

Reviewer #2: Partly

2. Has the statistical analysis been performed appropriately and rigorously? 

Reviewer #1: No

Reviewer #2: N/A

3. Have the authors made all data underlying the findings in their manuscript fully available?

Reviewer #1: Yes

Reviewer #2: Yes

4. Is the manuscript presented in an intelligible fashion and written in standard English?

Reviewer #1: No

Reviewer #2: Yes

5. Review Comments to the Author

Reviewer #1: This paper proposes a methodology for solving a portfolio optimization problem using a dynamic programming method. The manuscript has several structural drawbacks, some of which are as follows :

- The introduction section is not written in a proper condition. The contributions in this section are not adequately expressed. The difference between the proposed method and the previously published papers on applying the dynamic programming method to portfolio optimization problems is not discussed. Based on an author's claim, “The study aims to find a more efficient method for the portfolio optimization problems.” The reasons for the superiority of the proposed method should be considered in the introduction section.

- Based on constraint (3), eq(7) is incorrect.

- In the first presentation of the portfolio optimization problem, each constraint is numbered, but this is not the case in the second statement of the problem.

- The notation of eq(2) differed from the corresponding equation in Eq(7).

- The title “Dynamic programming” is a general topic that is unsuitable for a research paper.

- This paper needs a flowchart.

- The steps in algorithm 1 need a simple example to make following the method easier.

- The numerical result section should be improved critically. In the first step, different case studies should be defined clearly. The input parameters for each case are described in the tables, and appropriate references should be assigned to each section. Please make sure that all parameters are defined precisely. Finally, the results are discussed.

- The achieved results should be compared with other methods, especially the evolutionary methods.

Reviewer #2: This topic is interesting and seems to be practical. However, the quality of the manuscript should be improved carefully and based on the following comments.

1- The English and writing of the manuscript should be improved. It needs extensive editing; the choice of words, language, syntax, phrasing, punctuation should be thoroughly checked and revised.

2- To make the contribution of the paper more clear, authors should summarize the characteristics of their method in the section of the introduction.

3- The literature review sections should be improved; the focus should be on a critical analysis of the gradual advancement, as well as the current level, of the state-of-the-art, with quantitative information on the time & space complexity, as well as on the accuracy obtained by each cited methodology. The advancement offered by each cited methodology should be made clear.

4- Please provide the information of all indices in front of each mathematical equation. For example: Z, S, x

5- The comparative part of this paper is very weak. The author should make a more comparative study to compare the efficiency of the proposed techniques with well-known existing techniques. For this optimization problem, it is better to compare several algorithms.

6- Section 4 is not available!! (In section 4, two case studies of the US stock exchange are analyzed)

7- There are two sections 5 and in terms of numbering the sections of the article should be corrected.

8- To better understand the algorithm, it is recommended to use a flowchart with the algorithm.

6. PLOS authors have the option to publish the peer review history of their article (what does this mean?). If published, this will include your full peer review and any attached files.

Reviewer #1: No

Reviewer #2: **Yes: **Ali Mohammadzadeh

---

## [Decision Letter · Decision Letter 1]

8 Jul 2022

Constrained portfolio optimization with discrete variables: An algorithmic method based on dynamic programming

PONE-D-22-09775R1

Dear Dr. Jezeie,

We’re pleased to inform you that your manuscript has been judged scientifically suitable for publication and will be formally accepted for publication once it meets all outstanding technical requirements.

**Please note that there are some minor comments suggested by the second reviewer on the quality of equations and figures that can be addressed in the proof. **

Kind regards,

Seyedali Mirjalili

Academic Editor

PLOS ONE

Additional Editor Comments (optional):

Reviewers' comments:

Reviewer's Responses to Questions

**Comments to the Author**

1. If the authors have adequately addressed your comments raised in a previous round of review and you feel that this manuscript is now acceptable for publication, you may indicate that here to bypass the “Comments to the Author” section, enter your conflict of interest statement in the “Confidential to Editor” section, and submit your "Accept" recommendation.

Reviewer #1: All comments have been addressed

Reviewer #2: All comments have been addressed

2. Is the manuscript technically sound, and do the data support the conclusions?

Reviewer #1: Partly

Reviewer #2: Yes

3. Has the statistical analysis been performed appropriately and rigorously? 

Reviewer #1: N/A

Reviewer #2: Yes

4. Have the authors made all data underlying the findings in their manuscript fully available?

Reviewer #1: Yes

Reviewer #2: Yes

5. Is the manuscript presented in an intelligible fashion and written in standard English?

Reviewer #1: Yes

Reviewer #2: Yes

6. Review Comments to the Author

Reviewer #1: it is better to change the format of Tables 5 and 6.

please separate the column of firefly and GA from the proposed method

they have incorrect format

Reviewer #2: The authors could improve the quality of the manuscript such that I appreciate it. However, there are still some minor points to be taken into account.

1- Check the equation numbers (e.g., Equation 8).

2- Increase figure quality . (e.g., Remove the red underline from the images)

7. PLOS authors have the option to publish the peer review history of their article (what does this mean?). If published, this will include your full peer review and any attached files.

Reviewer #1: No

Reviewer #2: **Yes: **Ali Mohammadzadeh

---

## [Editor Report · Acceptance letter]

20 Jul 2022

PONE-D-22-09775R1 

Constrained portfolio optimization with discrete variables: An algorithmic method based on dynamic programming 

Dear Dr. Jezeie:

I'm pleased to inform you that your manuscript has been deemed suitable for publication in PLOS ONE. Congratulations! Your manuscript is now with our production department. 

Kind regards, 

on behalf of

Prof. Seyedali Mirjalili 

Academic Editor

PLOS ONE